# Highly Effective Fibrin Biopolymer Scaffold for Stem Cells Upgrading Bone Regeneration

**DOI:** 10.3390/ma13122747

**Published:** 2020-06-17

**Authors:** Camila Fernanda Zorzella Creste, Patrícia Rodrigues Orsi, Fernanda Cruz Landim-Alvarenga, Luis Antônio Justulin, Marjorie de Assis Golim, Benedito Barraviera, Rui Seabra Ferreira

**Affiliations:** 1Center for the Study of Venoms and Venomous Animals (CEVAP), UNESP—São Paulo State University, Botucatu 18610-307, Brazil; camilazorzella@hotmail.com (C.F.Z.C.); patriciabtu@yahoo.com.br (P.R.O.); bbviera@gnosis.com.br (B.B.); 2Botucatu Medical School, UNESP—São Paulo State University, Botucatu 18618-687, Brazil; marjorie.golim@unesp.br; 3College of Veterinary Medicine and Animal Husbandry (FMVZ), UNESP—São Paulo State University, Botucatu 18618-681, Brazil; fernanda@fmvz.unesp.br; 4Botucatu Biosciences Institute, UNESP—São Paulo State University, Botucatu 18618-689, Brazil; justulin@ibb.unesp.br

**Keywords:** bioproduct, biomaterial, scaffold, fibrin, stem cell

## Abstract

Fibrin scaffold fits as a provisional platform promoting cell migration and proliferation, angiogenesis, connective tissue formation and growth factors stimulation. We evaluated a unique heterologous fibrin biopolymer as scaffold to mesenchymal stem cells (MSCs) to treat a critical-size bone defect. Femurs of 27 rats were treated with fibrin biopolymer (FBP); FBP + MSCs; and FBP + MSC differentiated in bone lineage (MSC-D). Bone repair was evaluated 03, 21 and 42 days later by radiographic, histological and scanning electron microscopy (SEM) imaging. The FBP + MSC-D association was the most effective treatment, since newly formed Bone was more abundant and early matured in just 21 days. We concluded that FBP is an excellent scaffold for MSCs and also use of differentiated cells should be encouraged in regenerative therapy researches. The FBP ability to maintain viable MSCs at Bone defect site has modified inflammatory environment and accelerating their regeneration.

## 1. Introduction

Tissue repair is frequently necessary after skeletal diseases, congenital abnormalities, infections, trauma and surgical procedures after hematological, breast and ovary cancers. Fractures with bone loss often require grafts or implants. Autologous and allogeneic grafts represent about 90% of bone tissue transplants while inorganic matrices represent the other 10% [1,2]. Ideal implants must act as scaffold for bone regeneration with host tissue integration.

Main function of scaffolds is to offer structure and support for migration and specialization of different cells involved in healing. This structure should allow cell adhesion, attachment, differentiation, proliferation and biologic function for repair of the injured tissue [3].

Mesenchymal stem cells (MSCs) are used in tissue engineering [4,5,6] as an excellent alternative for bone repair since they are able to differentiate in osteoblasts as also in chondrocytes, myocytes, adipocytes and fibroblasts [7]. MSC applied in tissue repair has evolved progressively to improve or even substitute the healing capacity of bone tissue in partial or complete failure of the repair mechanism [8,9].

Combination of live cells with synthetic or natural scaffolds has been used to produce live tridimensional tissues that are functional, structural and mechanically identical to the original [10,11,12]. Different compounds have been used as scaffolds for MSCs [13] and can be classified as synthetic (i.e., hydroxyapatite and calcium triphosphate) [14] or biologic as fibrin biopolymers [15,16].

Synthetic osteoconductive implants have porous structures that promotes bone growth, however, the absence of an osteoinductive potential is still a limitation [17]. Fibrin matrix possesses some special characteristics that make it the scaffold of choice in tissue engineering [18]. Commercially available fibrin biopolymers are used in different surgical fields as hemostatic agents, healing promoters, cavity sealers and drug delivery in surgical sites [19,20]. Fibrin biopolymers have showed in vitro similar structure and mechanical properties to those of the fibrin clot in vivo [21,22].

Biocompatibility, biodegradability and the capacity to interact with MSC suggest that fibrin biopolymers are important vehicles for cell transplantation [20,21,23]. However, they are derived from human thrombin and fibrinogen that has a risk of infectious disease transmission and limited use due to possible lack of the main components [24,25,26].

Fibrin biopolymers commercially available today are produced from human thrombin and fibrinogen, being expensive and used only in specific surgical cases. Hence, this study evaluated a new fibrin biopolymer (FBP) composed of a mixture of a serine protease with thrombin-like enzyme activity, purified from *Crotalus durissus terrificus* snake venom and buffalo cryoprecipitate as a source of fibrinogen [27].

This new FBP has been used in experimental biomedical applications [28,29,30,31,32,33] such as nervous tissue [34,35] and bone repair [36] as also on the treatment of chronic venous ulcers in human patients [32,35]. In addition, the FBP enabled in vitro MSC adhesion, growth, had no negative effect on cell differentiation, and also maintained cell viability [15].

Although many associations of scaffolds and MSCs are being studied for bone defect healing there are still challenges to be faced [37,38,39,40]. Aiming to overcome current method limitations we evaluated the effect of this new FBP with MSCs and osteogenic differentiated MSCs on the treatment of critical-size defects in rats.

## 2. Material and Methods

### 2.1. Animals and Ethical Approval

All experiments were performed in 2-month-old male Wistar rats (*n* = 27) weighing between 200 and 250 g. Animals were housed in polycarbonate cages (4 per cage) and were kept at 21 ± 2 °C under a 12-h light/dark cycle and a humidity of 60% ± 10%. The animals had ad libitum access to food pellets of standard rodent diet and water. The Experimental ethics committee for the protection of experimental animal welfare of Botucatu Medical School, Sao Paulo State University, Brazil has approved this study (No. 968-12). The guidelines of the European convention for the protection of vertebrate animals used for experimental purposes and, the Guide for the care and use of laboratory animals and good laboratory practices were fully adopted.

### 2.2. Fibrin Biopolymer (FBP)

The FBP was kindly provided by center for the Study of Venoms and Venomous Animals (CEVAP), Brazil. Components were distributed in three vials containing thrombin-like enzyme, animal cryoprecipitate and diluent and were kept frozen at −20 °C until use [35,41,42,43,44]. At time of surgery, contents were immediately mixed according to the manufacturer’s package insert.

### 2.3. Cell Isolation and Culture

Twelve 10-day-old Wistar rats were euthanized with halothane overdose (MAC > 5%) and used as bone marrow donors. Stem cells were harvested by washing of femur marrow cavity with the injection of Dulbecco’s modified Eagle’s medium (DMEM) (Gibco Laboratories, Grand Island, NE, USA).

The material was pooled, centrifuged at 2000 rpm for 10 min and resuspended in complete culture medium composed of DMEM (Gibco Laboratories) supplemented with 20% fetal bovine serum (Sigma-Aldrich, St. Louis, MO, USA), 100 µg/mL of penicillin/streptomycin solution (Gibco Laboratories) and 3 µg/mL of amphotericin B (Gibco Laboratories).

Cells were seeded in 75 cm^2^ culture flasks and placed in a 5% CO_2_ incubator at 37.5 °C. Culture medium was changed every 3 days and cell growth and adherence were monitored by inverted microscopy. Cells were subcultured when reached 80% confluence. All experiments were performed with MSCs at passage 3 (P3). To perform the passage, culture medium was discarded; the cells were washed with 2 mL of PBS followed by addition of Tryple Select (Gibco Laboratories) for cell trypsinization and the flask was maintained in an incubator oven for 5 min.

These were centrifuged for 10 min at 2000 rpm and resuspended in culture media. Cells were counted and 1 × 10^6^ cells/dose were used in association with FBP for the treatment of the bone defect throughout the experiment [35].

Cells were characterized by flow cytometry (FACS Calibur; BD Pharmingen, San Diego, CA, USA) using monoclonal antibodies for specific positive and negative markers (Table 1) [13,14,45,46]. Assays were performed using 2 × 10^5^ cells and data were analyzed using the Cell Quest Pro software after acquisition of 20,000 events. Functional characterization was also performed as cells were differentiated in osteogenic, chondrogenic and adipogenic lineages after the third passage [22,36,47].

### 2.4. Osteogenic Differentiation of MSCs

After cell culture had reached 70% confluence, culture medium was replaced by Stem Pro Osteogenesis Differentiation Kit medium (Gibco Life Technologies A10072-01, Carlsbad, CA, USA), composed of 73% osteocyte/chondrocyte differentiation basal medium (Gibco Life Technologies A10069-01, Carlsbad, CA, USA), 5% osteogenesis supplement (Gibco Life Technologies A10066-01, Carlsbad, CA, USA), 1% penicillin/streptomycin, 1% amphotericin B and 20% fetal bovine serum (Sigma-Aldrich, St. Louis, MO, USA). The differentiation medium was replaced every 3 days for 12 days.

Then, cells were fixed in ice-cold 70% ethanol, washed in distilled water and stained in 2 mL of alizarin red (Invitrogen Life Science Technologies, Carlsbad, CA, USA) for 30 min at room temperature. After the dye was removed, cells were washed four times in distilled water and observed in an inverted light microscope [17,48].

### 2.5. Animals and Surgical Protocols

Animals were weighed and anesthetized with ketamine solution (1 mL/kg) and xylazine hydrochloride (0.25 mL/kg) intraperitoneally. Cross sections of the thigh through the upper- and middle-third of the femur allowed a critical defect of 5 mm to be performed on the distal epiphysis of the right femur with a low rotation drill (Beltec) under constant irrigation of 0.9% sterile saline to prevent overheating [49]. Postoperative analgesia with intramuscular flunixin-meglumine (1 mg/kg) was performed every 24 h for three days.

Animals were distributed in three experimental groups of 9 animals each: (FBP), the animals were treated with fibrin biopolymer only; (FBP + MSCs) treated with fibrin biopolymer in association with mesenchymal stem cells; and (FBP + MSC-D) treated with fibrin biopolymer in association with differentiated mesenchymal stem cells.

Three untreated animals were used as control to assess critical defects throughout the experimental period and evaluated radiographically at 42 dpi.

Cells were mixed in 100 µL of FBP immediately before injection at 1 × 10^6^ cells/dose for FBP + MSCs and FBP + MSC-D groups. Surgeries were carried out under sterile conditions.

### 2.6. Radiographic Evaluation

Radiographic imaging of the rat femurs was conducted at 3rd, 21st and 42nd days using a digital GE model E7843X system (GE Healthcare, Chicago, IL, USA).

### 2.7. Histological Analysis

Femurs were collected and fixed in 10% buffered formalin for 24 h at 4 °C and were decalcified with 10% neutralized EDTA (Sigma) for 4 weeks; then dehydrated with an ascending series of ethanol concentrations, cleared in xylene and embedded in Paraplast (Sigma). Histological sections (6 µm) were stained with hematoxylin-eosin (H&E) for general morphologic analysis or picrosirius for collagen fibers (type I and type III) quantification and stereological analysis [50]. The color displayed under polarizing microscopy was a result of fiber thickness, as well as the arrangement and packing of the collagen molecules. Normal tightly packed thick collagen fibers had polarization colors in the red spectrum while thin or unpacked fibers had green birefringence [51]. Sections were observed under normal and polarized light, and digitalized images were analyzed using Leica Q-win software (Version 3.0) to calculate mean collagen fiber area.

Non-injured bone was used to show differences with our injured groups in radiographic evaluation and histological analysis (H&E and picrosirius).

### 2.8. Scanning Electron Microscopy (SEM)

SEM analyses were performed using a Quanta 200 electron microscope (FEI Company, Hillsboro, OR, USA). Bone samples were fixed in 2.5% glutaraldehyde in 0.1-M PBS pH 7.3 for 4 h. Samples were then removed and washed three times for 5 min in distilled water. Subsequently, samples were immersed for approximately 40 min in 0.5% osmium tetroxide and washed three times in distilled water; dehydrated in increasing concentrations of ethanol (7.5% to 100%); dried in a critical point apparatus with liquid carbon dioxide, mounted on appropriate chucks, metallized and gold-coated [37].

## 3. Results

### 3.1. MSCs Expansion and Characterization

MSCs exhibited fibroblastoid morphology (Figure 1A). Cells remained in primary culture until reached 80% confluence after approximately 07 days; then subcultured up to the third passage for use. Flow cytometry showed that 97.57%, 98.49%, 84.47% and 91.70% of the cells expressed positive markers ICAM-I, CD90, CD73 and CD44, respectively (Figure 1B–E). Negative markers MHC II, CD34, CD45, RT-1 and CD11b were expressed by 1.45%, 1.32%, 2.39%, 1.80% and 1.74% of cells, respectively (Figure 1F–J). These results demonstrate that cultured cells exhibited the characteristic phenotype of MSCs.

### 3.2. MSCs Osteogenic Differentiation

Figure 1A (in detail) also shows calcium deposits observed in MSC cultures after 12 days of incubation in specific differentiation media. Mineral deposits were detected by presence of red staining on the extracellular medium, thus confirming the MSC osteogenic differentiation.

### 3.3. Radiographic Evaluation

Radiographic analyses (Figure 2) showed that defects were evident in all groups 3 days after surgical procedure. At the 21st day, FBP + MSC-D treated group presented efficient healing as the defect was almost completely filled. At day 42, FBP + MSC-D group showed total bone healing and it was possible to observe improvement of repair on FBP + MSC treated group. The control group (non-treated) showed that the bone defect performed was critical and did not heal at 42 days post intervention (Figure 2).

### 3.4. Scanning Electron Microscopy (SEM)

Scanning electron microscopy imaging evidenced the bone structure at injury site. On the 3rd day after surgery defect was evident in all groups. Group treated with FBP + MSC-D showed markedly higher injury repair when compared to the other two groups at day 21. After 42 days it was possible to observe bone tissue deposits in all treated groups. However, in groups FBP and FBP + MSCs the defect has not been completely repaired as could be observed on group FBP + MSC-D (Figure 3).

### 3.5. Histological Analysis

H&E stained materials are demonstrated in Figure 4. A progressive bone matrix deposition was observed during the experimental period. Presence of a fibrillary material, similar to FBP structure, inside the defect 3 days after surgery on the group treated only with FBP evidences that it has adhered to injury site. Bone fragments probably from the surgical procedure were also observed adhered to the fibers. There was a significant increase in cellularity associated to the biomaterial. On the group treated with FBP + MSCs the presence of newly formed trabecular bone on the defect margins was evident after 21 days as well as in the FBP + MSC-D treated group. At day 42, from histological perspective, all defects were partially repaired, although in the FBP + MSC-D treated group newly formed bone was more abundant and its structure more similar to normal mature bone tissue.

Figure 5 shows collagen fibers formation through picrosirius staining under polarized light. Yellow-reddish staining represents mature thick fibers, as demonstrated by the FBP group, while green staining shows recent synthesized and immature fibers. On the first 3 days there was no evidence of collagen formation in all groups. After 21 days, there were observed thin immature green fibers within thick yellow and red mature collagen showing an increase in collagen synthesis in all three groups. In addition, there were also a high number of cells adhered to the scaffold claiming that MSCs injected with the FBP remained at injury site and have differentiated for matrix synthesis. Our results show that even after 21 days cells were associated with the fibrin structure strengthening the use of FBP as a scaffold for cell delivery. The collagen synthesis pattern was similar in FBP and FBP + MSC groups at day 42, but it was higher on group FBP + MSC-D.

## 4. Discussion

Although autologous bone graft remains the gold standard for healing large bone defects, grafting procedure complexity increases due to donor site morbidity, increased risk of infection and poor ability to fill complex defects [52], besides the feasibility to obtain material in adequate quantity and quality. However, the auto graft has its limitations, including donor-site morbidity and supply limitations, hindering this as an option for bone repair [53].

Delivery systems for MSCs and evaluation of their safety and effectiveness also need to be investigated [54]. Scaffolds for bone tissue repair must induce bone formation and provide a suitable microenvironment for growth of bone cells exhibiting osteoconductivity, osteogenicity and osteoinductivity [42].

Our results have showed the fibrin biopolymer (FBP) scaffold potential for MSCs in bone-in vivo repair and its biocompatibility. Association between FBP and MSC-D was able to promote total repair in critical size defect in rat femurs in almost half-time when compared to other studied treatments.

Commercially available fibrin biopolymers, also called fibrin sealants, consist of human fibrinogen and thrombin. The FBP used in this study is composed of a mixture of a serine protease with thrombin-like enzyme activity, purified from *Crotalus durissus terrificus* snake venom and buffalo cryoprecipitate as a source of fibrinogen [30,40].

Previous studies with FBP scaffold have shown no cytotoxicity condition for MSCs [17,35,55,56]. Furthermore, have shown that FBP promotes chemotaxis for M2 macrophages producing anti-inflammatory profile and neoangiogenesis [32]. We did not observe signs of local inflammation proved by animals’ postoperative status with normal cicatrization and absence of phlogistic signs of inflammation and surgical site infection such as erythema, local edema or exudates. In addition, there were few leukocyte infiltrates that are characteristics of foreign body reactions evidencing FBP biocompatibility.

Spejo et al. [56] showed the use of FBP in animals models increased influx of macrophages after 3 and 7 days after injury due to gene expression increase of M1 and M2 macrophage markers and anti-inflammatory and pro-inflammatory cytokines as seen by qRT-PCR. The authors hypothesize that the fibrinolysis process can change the local environment generating a predominantly proinflammatory milieu in the first moments of healing.

Gasparotto et al. [16] have demonstrated in vitro interactions of the FBP with MSCs either in scanning electron microscopy (SEM) or in transmission electron microscopy (TEM). Authors concluded that FBP showed ideal plasticity and MSCs homing without differentiation effects. Orsi et al. [35] have evaluated the effect of FBP associated with both MSC and MSC-D on osteoporotic female rats and showed that the association promotes a higher bone formation compared to the control group after 14 days. They also have demonstrated that there was no cytotoxicity of FBP for MSCs.

Flow cytometry (CF) proved to be effective for the MSCs characterization. Cells presented expected fusiform shape in culture and FC panel chosen was adequate and agreed with other authors that stated MSCs should present positive for CD73, CD90, CD105 e ICAM and negative for CD45, CD34, CD14 or CD11b, CD79 or CD19 [50,57,58,59]. Additionally, rat bone marrow derived MSCs have differentiated in osteogenic lineage after 12 days on presence of specific differentiation media corroborating Vilquin & Rosset [9].

FBP helped cicatricial evolution with total wound healing after observational period. Group treated with FBP and MSC-D highlighted from the others as it presented complete repair after 21 days. Xu et al. [49] also evaluated a new scaffold composed by BG-COL-HYA-OS and MSCs in rat femur regeneration and have observed a significant injury filling after 42 days.

FBP has also been used as a scaffold in the regeneration of other tissues. Association between MSCs and fibrin scaffold for regenerative process after peripheral nerve tubulization has improved nerve regeneration by positively modulating the reactivity of Schwann cells [33].

MSC therapy when associated with a FBP act as neuroprotective and shifts the immune response to a proinflammatory profile due FBP kept EGFP-MSCs at the glial scar region in the ventral funiculus after 28 days [56].

Radiographic analysis is an auxiliary measure for repair evaluation in bone lesions as it provides neither information about bone quality in new tissue nor it allows for a clear visualization of old-new bone interface [60]. Strategies to stimulate and reinforce the mobilization and homing of MSCs have become a key point in regenerative medicine [61]. Histological and SEM analysis confirmed radiographic findings and also complemented the information.

In the control sample, the histological images represented areas of mature cortical bone, composed of mineralized collagen fibers stacked parallel to form lamellae. Collagen fibers were made up of closely packed thick fibrils and exhibited an intense birefringence of yellow/red color under the polarizing microscope.

In the experimental group that received fibrin biopolymer and differentiated mesenchymal stem cells—despite the formation of new bone faster than the other groups—bone regeneration was not mature. The bone matrix consists of loosely arranged thin collagen fibrils, which exhibited a weak birefringence of green color interconnected to the thick yellow fibers under the polarizing microscope. This result was consistent with the timing of regeneration of different bone tissue (cortical and cancellous). In cortical bone, the remodeling process takes twice as long to remodel than cancellous bone [62].

Considering the three analysed allowed us to conclude that the association between FBP and MSC-D was able to promote total repair in critical size defect in rat femur and shortened bone repair compared to other evaluated treatments.

We know that bone marrow-derived MSCs are a better choice for bone engineering than other MSC sources due to the greater potential for chondrogenic differentiation [63]. However, the way in which MSCs harbor the lesion site is not yet clear, however the chemoattracting molecules released at the bone lesion site should play an essential role in attracting MSCs [64]. All of this indicates that the MSCs are dependent on the attractor/receiver [65]. However, the downside of the return property of MSCs is that they can harbor other tissues, even if they develop tumors [66,67] or suffer necrosis/apoptosis, which is very harmful. Hence, a scaffold that allows to maintain, as viable MSCs at the site of the bone injury should always be considered.

## 5. Conclusions

The recruitment and homing of MSCs are essential for bone healing. MSC mobilization accelerates bone healing mainly by stimulating angiogenesis and coordinating bone remodeling. FBP presented as a highly effective scaffold for applications in bone lesions because it accelerated tissue regeneration. We have concluded that the use of fibrin scaffold for mesenchymal stem cells pre differentiated in bone lineage have accelerated the bone healing process by keep cells viable on injury site without any adverse events.

## Figures and Tables

**Figure 1 materials-13-02747-f001:**
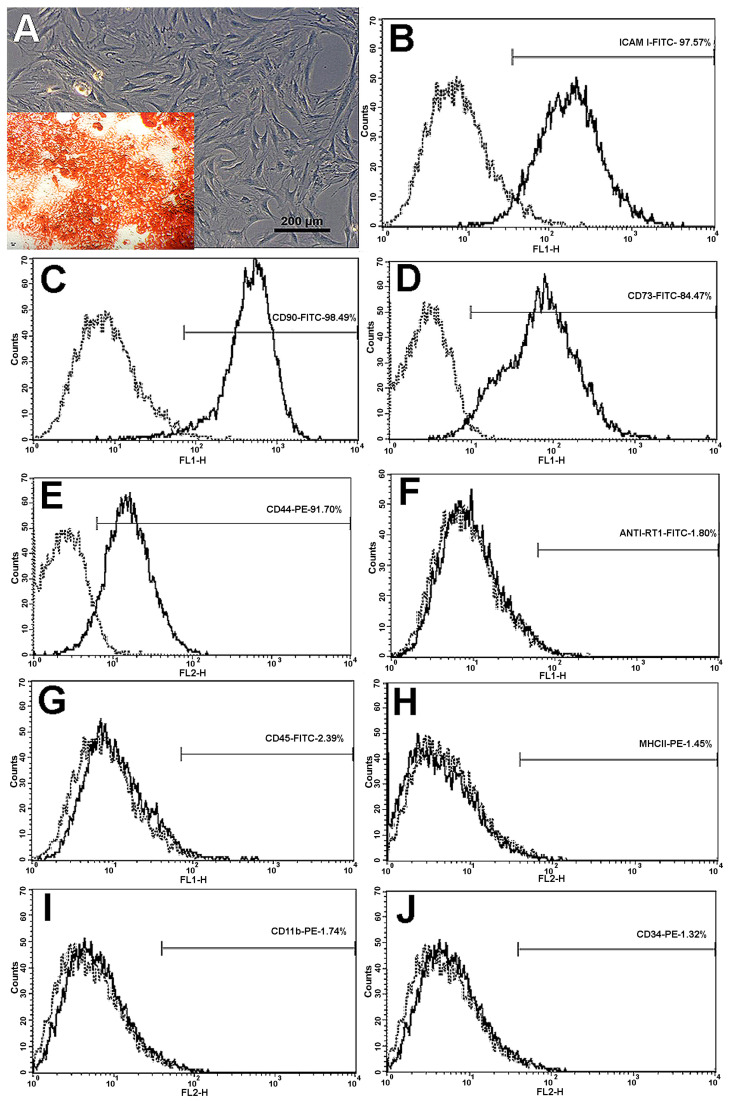
(**A**) Cultivated rat MSCs showing expected fibroblastoid (fusiform) shape. In detail: calcium deposits stained red in MSC cultures after 12 days of differentiation. (**B**) ICAM-I; (**C**) CD90; (**D**) CD73; (**E**) CD44; (**F**) anti-RT1; (**G**) CD45; (**H**) MHC II; (**I**) CD11b; (**J**) CD34.

**Figure 2 materials-13-02747-f002:**
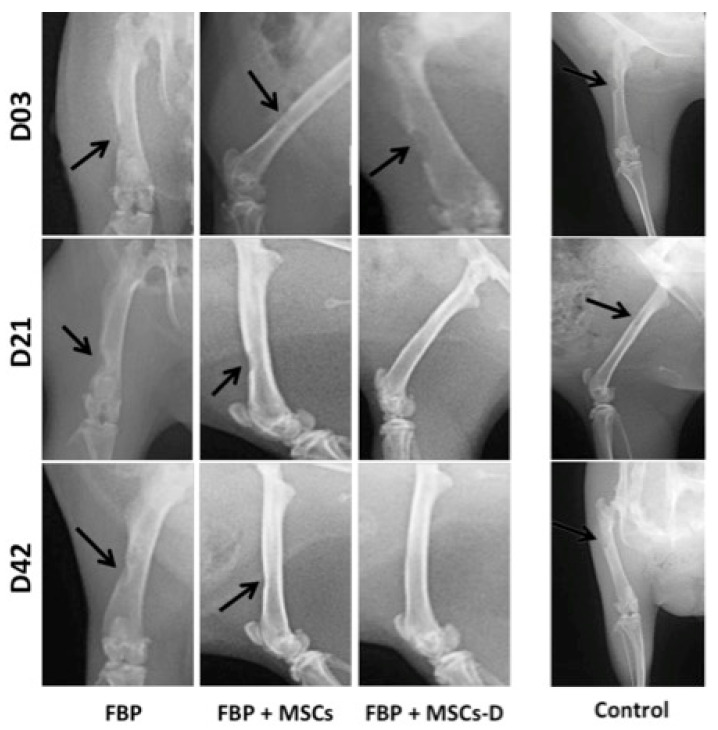
Radiographic analysis of bone injury in femur of rats at 3, 21 and 42 days post injury. FBP (fibrin biopolymer only); FBP + MSCs (fibrin biopolymer + mesenchymal stem cells); FBP + MSC-D (fibrin biopolymer + differentiated mesenchymal stem cells). Control shows non-treated bone for comparison.

**Figure 3 materials-13-02747-f003:**
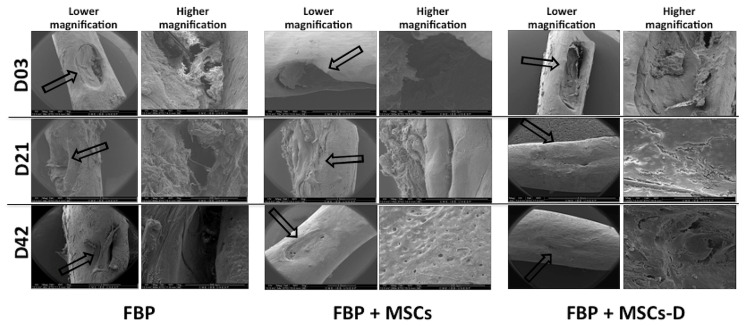
Scanning electron microscopy (SEM) imaging of bone injury in femur of rats at 3 (D03), 21 (D21) and 42 (D42) days post injury. FBP (fibrin biopolymer only); FBP + MSCs (fibrin biopolymer + mesenchymal stem cells); FBP + MSC-D (fibrin biopolymer + differentiated mesenchymal stem cells). Lower magnification (40×) and higher magnification (280×). Black arrow shows the injury area.

**Figure 4 materials-13-02747-f004:**
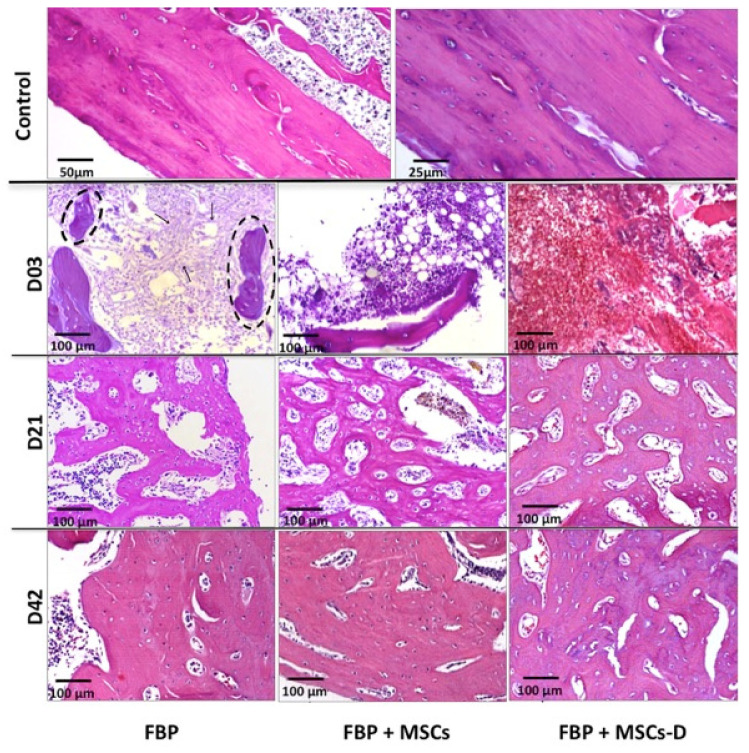
Histological analysis of bone regeneration tissue in femur of rats at 3 (D03), 21 (D21) and 42 (D42) days post injury stained with H&E. FBP (fibrin biopolymer only); FBP + MSCs (fibrin biopolymer + mesenchymal stem cells); FBP + MSC-D (fibrin biopolymer + differentiated mesenchymal stem cells); Control show a non-injured bone for comparison. Arrows: fibrillary material; dashed circle: bone fragments.

**Figure 5 materials-13-02747-f005:**
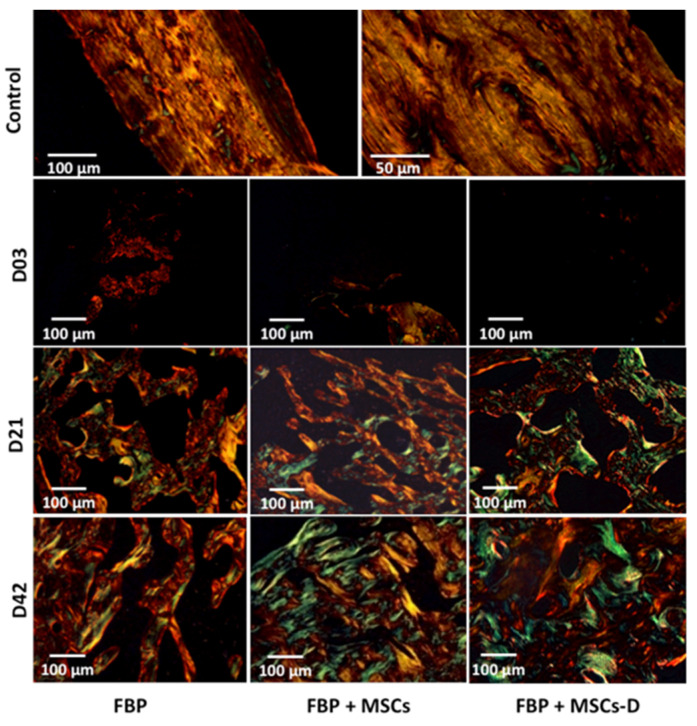
Picrosirius staining under polarized light showing collagen formation in injury filling in femur of rats at 3, 21 and 42 days post injury. FBP (fibrin biopolymer only); FBP + MSCs (fibrin biopolymer + mesenchymal stem cells); FBP + MSC-D (fibrin biopolymer + differentiated mesenchymal stem cells). Control shows a non-injured bone for comparison.

**Table 1 materials-13-02747-t001:** Surface markers for mesenchymal stem cells (MSCs) characterization.

**Negative Markers**	
RT1	anti-RT1-Aw2-FITC, clone MRC OX-18; Abcam, Cambridge, MA, USA
CD34	anti-CD34-PE, clone ICO-115; Abcam, Cambridge, MA, USA
CD11b	anti-CD11b-PE, clone ED8; Abcam, Cambridge, MA, USA
CD45	anti-CD45-FITC, clone MRC OX-1; Abcam, Cambridge, MA, USA
MHCII	anti-rat MHC CLASS II RT1D-PE, clone MRC OX-17; Abcam, Cambridge, MA, USA
**Positive Markers**	
CD73	purified mouse anti-rat CD73; clone 5F/B9, BD Pharmingen, San Diego, CA, USA
CD90	anti-CD90/Thy1-FITC, clone FITC.MRC OX-7; Abcam, Cambridge, MA, USA
CD44	anti-CD44-PE, clone OX-50; Abcam, Cambridge, MA, USA
ICAM-I	anti-ICAM-I-FITC, clone 1A29; Abcam, Cambridge, MA, USA

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
