# Peer review of "Highly Effective Fibrin Biopolymer Scaffold for Stem Cells Upgrading Bone Regeneration"

_materials, 2020, doi:10.3390/ma13122747_

Round 1

Reviewer 1 Report

The study entitled 'Highly Effective Fibrin Biopolymer Scaffold for Stem Cells Upgrading Bone Regeneration' aims to prepare FBP scaffolds loaded with MSCs for bone regeneration. Not surprisingly FBP+MSC-D association shows the most effective treatment. The results are presented well, however,  the following comments should be addressed:

  1. Grammatical mistakes:

Line 22-25: The sentence “Upgrade on…was…enable…cytokines” needs to be grammatically revised and make it clearer.

Line 32: “…inorganic matrices represents…” should be changed to “…inorganic matrices represent…”.

Line 49: The sentence “…can used in…are…” is incorrect.

Line 76: “…under a 12 h light/dark cycle a humidity of…” need to be revised as “…under a 12 h light/dark cycle and a humidity of…”.

Line 101: The sentence “Thus were…” needs to be grammatically revised.

Line 183: The word “improve” need to be revised as “improvement”.

Line 203: “H&E stained material are…” should be revised as “H&E stained materials are…”.

Line 239: “Scaffolds to bone tissue repair…” should be rewritten as “Scaffolds for bone tissue repair…”.

Line 242-243: The sentence “Our results demonstrate…evidencing its biocompatibility.” needs to be grammatically revised.

Line 251: The sentence “Also have shown that…” needs to be grammatically revised.

Line 283: “MSCs therapy…act as…” should be revised as “MSC therapy…acts as…”.

Line 288: The sentence “According…to stimulate and reinforce…” needs to be grammatically revised

  1. Spelling mistakes and clerical errors:

Line 41: The word “condrocytes” is incorrect.

Line 86: “…surgery contents were…” should be revised as “…surgery, contents were…”.

Line 122, 123 and 126: “Kg” should be replaced by “kg”.

Line 137: “42th” should be revised as “42nd

Line 145: In the sentence “The color…microscopy, is …”, the comma is redundant.

Line 162: In the sentence “MSCs… (Figure 1A) Cells…”, a period is missing.

Line 173: In the sentence “(F, G, J, I, J). Negative…”, the period is redundant.

Line 249: “[40,30]” should be revised as “[30,40]”.

Line 254: The word “flogistic” is incorrect.

Line 257, 262 and 279: The commas are redundant. Line 262 says "Error! ..."

  1. There should be a space between every number and its unit.
  2. Line 189: Please make sure whether the control shows a non-injured bone or a non-treated bone. Images show injury.
  3. Line 225-226: The logic of this sentence “The same…, thus it…” is incorrect.
  4. Line 250: The sentence “…no cytotoxicity becoming an excellent scaffold for MSC” should be more clear.
  5. Line 269: The sentence “MSC characterization was effective either in microscopy or in flow cytometry (FC).” should be more clear.
  6. Line 279-282: The sentence “Cartarozzi et al. have...cells” should be more clear.
  7. In discussion part, the authors should give more analyses of the results, such as causes and explanations, rather than describing backgrounds. In addition, the authors provide many results of others’ researches. What are the differences and relations between others’ and the authors’ works? What inspirations did the authors get from others’ work?
  8. In the conclusions part, the sentence “We have…events, accelerate…healing process.” (line 297-299) is complicated and it should be more clear. There are also some grammatical mistakes.
  9. I would advise using a different color/style for the arrows in Figure 3 as they are hardly visible.

Author Response

Dear reviewer. Thank you very much for all the considerations. We are answering each of them below.

REVIEW REPORT 1

Comments and Suggestions for Authors

The study entitled 'Highly Effective Fibrin Biopolymer Scaffold for Stem Cells Upgrading Bone Regeneration' aims to prepare FBP scaffolds loaded with MSCs for bone regeneration. Not surprisingly FBP+MSC-D association shows the most effective treatment. The results are presented well, however, the following comments should be addressed:

  1. Grammatical mistakes:

Line 22-25: The sentence “Upgrade on…was…enable…cytokines” needs to be grammatically revised and make it clearer.

R: The sentence was changed on the text by:

“The FBP ability to maintain viable MSCs at bone defect site has modified inflammatory environment and accelerating their regeneration.”

Line 32: “…inorganic matrices represents…” should be changed to “…inorganic matrices represent…”.

R: The sentence was changed.

Line 49: The sentence “…can used in…are…” is incorrect.

R: The sentence was changed on the text to:

“Fibrin matrix possesses some special characteristics that make it the scaffold of choice in tissue engineering.”

Line 76: “…under a 12 h light/dark cycle a humidity of…” need to be revised as “…under a 12 h light/dark cycle and a humidity of…”.

R: The sentence was revised.

“Animals were housed in polycarbonate cages (4 per cage) and were kept at 21 ± 2 °C under a 12h light/dark cycle and a humidity of 60± 10%.”

Line 101: The sentence “Thus were…” needs to be grammatically revised.

The sentence was revised.

“Thus were centrifuged for 10 minutes at 2000 rpm and re-suspended in culture media. Cells were counted and 1x106 cells/dose were used in association with FBP for the treatment of the bone defect throughout the experiment.”

Line 183: The word “improve” need to be revised as “improvement”.

The sentence was revised.

“At day 42, FBP + MSC-D group showed total bone healing and it was possible to observe improvement of repair on FBP + MSC treated group.”

Line 203: “H&E stained material are…” should be revised as “H&E stained materials are…”.

The sentence was revised.

“H&E stained materials are demonstrated in Figure 4.”

Line 239: “Scaffolds to bone tissue repair…” should be rewritten as “Scaffolds for bone tissue repair…”.

The sentence was revised.

“Scaffolds for bone tissue repair must induce bone formation and provide a suitable microenvironment for growth of bone cells exhibiting osteoconductivity, osteogenicity and osteoinductivity.

Line 242-243: The sentence “Our results demonstrate…evidencing its biocompatibility.” needs to be grammatically revised.

The sentence was revised.

“Our results have showed the Fibrin Biopolymer (FBP) scaffold potential for MSCs in bone-in vivo repair and its biocompatibility.”

Line 251: The sentence “Also have shown that…” needs to be grammatically revised.

The sentence was revised.

Also have shown that FBP promotes chemotaxis for M2 macrophages producing anti-inflammatory profile and neoangiogenesis.

Line 283: “MSCs therapy…act as…” should be revised as “MSC therapy…acts as…”.

The sentence was revised.

“MSC therapy when associated with a FBP act as neuroprotective and shifts the immune response to a proinflammatory profile due FBP kept EGFP-MSCs at the glial scar region in the ventral funiculus after 28 days.”

Line 288: The sentence “According…to stimulate and reinforce…” needs to be grammatically revised.

The sentence was revised.

“Strategies to stimulate and reinforce the mobilization and homing of MSCs have become a key point in regenerative medicine [61].”

Spelling mistakes and clerical errors:

Line 41: The word “condrocytes” is incorrect.

Line 86: “…surgery contents were…” should be revised as “…surgery, contents were…”.

Line 122, 123 and 126: “Kg” should be replaced by “kg”.

Line 137: “42th” should be revised as “42nd

Line 145: In the sentence “The color…microscopy, is …”, the comma is redundant.

Line 162: In the sentence “MSCs… (Figure 1A) Cells…”, a period is missing.

Line 173: In the sentence “(F, G, J, I, J). Negative…”, the period is redundant.

Line 249: “[40,30]” should be revised as “[30,40]”.

Line 254: The word “flogistic” is incorrect.

Line 257, 262 and 279: The commas are redundant. Line 262 says "Error! ..."

There should be a space between every number and its unit.

The all Spelling mistakes and clerical errors were corrected

Line 189: Please make sure whether the control shows a non-injured bone or a non-treated bone. Images show injury.

The sentence was corrected:

“Control shows a non-treated bone as comparative.”

Line 225-226: The logic of this sentence “The same…, thus it…” is incorrect.

The sentence was changed:

“The collagen synthesis pattern was similar in FBP and FBP + MSC groups at day 42, but it was higher on group FBP + MSC-D.”

Line 250: The sentence “…no cytotoxicity becoming an excellent scaffold for MSC” should be more clear.

The sentence was changed:

“Previous studies with FBP scaffold have shown no cytotoxicity condition for MSCs.”

Line 269: The sentence “MSC characterization was effective either in microscopy or in flow cytometry (FC).” should be more clear.

The sentence was changed:

“Flow cytometry (CF) proved to be effective for the MSCs characterization.”

Line 279-282: The sentence “Cartarozzi et al. have...cells” should be more clear.

The sentence was changed:

“Association between MSCs and fibrin scaffold for regenerative process after peripheral nerve tubulization has improved nerve regeneration by positively modulating the reactivity of Schwann cells [33].”

In discussion part, the authors should give more analyses of the results, such as causes and explanations, rather than describing backgrounds. In addition, the authors provide many results of others’ researches. What are the differences and relations between others’ and the authors’ works? What inspirations did the authors get from others’ work?

We believe that the discussion chapter contains articles that are references for a better understanding of the manuscript. In this case, the use of pre-differentiated stem cells proved to be more effective in bone regeneration. This has not been done so far, although the use of fibrin as a scaffold for cells has been widely studied. Thus, the uses of our fibrin scaffold demonstrates the possibility of its use for different patterns of cells with the purpose o apply in regenerative engineering.

In the conclusions part, the sentence “We have…events, accelerate…healing process.” (line 297-299) is complicated and it should be more clear. There are also some grammatical mistakes.

The sentence was changed:

“We have concluded that the use of fibrin scaffold for mesenchymal stem cells pre differentiated in bone lineage have accelerated the bone healing process by keep cells viable on injury site without any adverse events.”

I would advise using a different color/style for the arrows in Figure 3 as they are hardly visible.

The arrows colors of Figure 3 were reinforced.

Reviewer 2 Report

The authors demonstrate demonstrated the bone repair by using fibrin biopolymer (FBP) as a scaffold and Mesenchymal Stem Cell differentiated in boon lineage (MSC-D). Compared with the treatment with FBP, FBP+MSC-D showed clearly earlier bone repair, which was proved by radiographic, SEM, Histological analysis and Picrosirius staining images. This paper involved clear and strong evidences to prove the effect of FBP and MSC-D on the bone repair. Thus, this paper should be published after some revisions.

1. Line 67-69

Were there reasons the authors selected the current component of fibrin biopolymer explained by “The FBP used in this study is composed of a mixture of a serine protease with thrombin-like enzyme activity, purified from Crotalus durissus terrificus snake venom and buffaloes cryoprecipitate as a source of fibrinogen”, from various commercially available candidates? They should explained in main text.

2. Section 3.5 and Figure 4

The authors had better add arrows to Figure to explain a fibrillary material, bone fragments and trabecular bone.

3. Section 3.5 and Figures 4 & 5

According to images of control sample in Figure 4 and 5, bone was anisotropic aligned. However, in the FBP+MSC-D treatment, bone cells appeared to be randomly oriented with no orientation. The addition of discuss about it may improve this paper.

4.

How about the comparison with MSC-D treatment without FBP? If the authors have not done it, they should compare the result of FBP-MSC-D with that of MSC-D reported previously in Discussion section because the main aim of this paper is the effect of FBP scaffold.

5.

The authors had better explain relationship and differences between this paper and previous papers such as [16] and [35]. How position was this paper in your works? What was new?

Author Response

Dear reviewer. Thank you very much for all the considerations. We are answering each of them below.

REVIEW REPORT FORM 2

Comments and Suggestions for Authors

The authors demonstrate demonstrated the bone repair by using fibrin biopolymer (FBP) as a scaffold and Mesenchymal Stem Cell differentiated in boon lineage (MSC-D). Compared with the treatment with FBP, FBP+MSC-D showed clearly earlier bone repair, which was proved by radiographic, SEM, Histological analysis and Picrosirius staining images. This paper involved clear and strong evidences to prove the effect of FBP and MSC-D on the bone repair. Thus, this paper should be published after some revisions.

  1. Line 67-69

Were there reasons the authors selected the current component of fibrin biopolymer explained by “The FBP used in this study is composed of a mixture of a serine protease with thrombin-like enzyme activity, purified from Crotalus durissus terrificus snake venom and buffaloes cryoprecipitate as a source of fibrinogen”, from various commercially available candidates? They should explained in main text.

R: Thank you for the comment. We have added the sentence.

“Fibrin biopolymers commercially available today are produced from human thrombin and fibrinogen, being expensive and used only in specific surgical cases. So, this study evaluated a new fibrin biopolymer (FBP) composed of a mixture of a serine protease with thrombin-like enzyme activity, purified from Crotalus durissus terrificus snake venom and buffaloes cryoprecipitate as a source of fibrinogen [27].”

  1. Section 3.5 and Figure 4

The authors had better add arrows to Figure to explain a fibrillary material, bone fragments and trabecular bone.

R: Arrows and dashed circle were added on Figure 4 to show fibrillary material and bone fragments.

  1. Section 3.5 and Figures 4 & 5

According to images of control sample in Figure 4 and 5, bone was anisotropic aligned. However, in the FBP+MSC-D treatment, bone cells appeared to be randomly oriented with no orientation. The addition of discuss about it may improve this paper.

R: Thank you for the comment. The sentence was added on the discussion section:

“In the experimental group that received Fibrin Biopolymer and Differentiated Mesenchymal Stem Cells, despite the formation of new bone faster than the other groups, bone regeneration is not mature. The bone matrix consists of loosely arranged thin collagen fibrils, which exhibited a weak birefringence of green color interconnected to the thick yellow fibers under the polarizing microscope. This result is consistent with the timing of regeneration of different bone tissue (cortical and cancellous). In cortical bone, the remodeling process takes twice as long to remodel than cancellous bone [62].”

  1. How about the comparison with MSC-D treatment without FBP? If the authors have not done it, they should compare the result of FBP-MSC-D with that of MSC-D reported previously in Discussion section because the main aim of this paper is the effect of FBP scaffold.

R: The treatment of bone wounds is already widely explored in the literature and its use in isolation presents several problems. To get around this situation, several compounds and scaffolds are mixed into cells in order to maintain viable pathways for a longer time at the injury site and to prevent cells from wishing and circulating freely throughout the rest of the body.

In this way, on the discussion section we have added a sentence below to make this clear to the reader:

“We know that bone marrow-derived MSCs are a better choice for bone engineering than other MSC sources due to the greater potential for chondrogenic differentiation [63]. However, the way in which MSCs harbor the lesion site is not yet clear, however the chemoattracting molecules released at the bone lesion site should play an essential role in attracting MSCs [64]. All of this indicates that the MSCs are dependent on the attractor / receiver [65]. However, the downside of the return property of MSCs is that they can harbor other tissues, even if they develop tumors [66, 67], or suffer necrosis / apoptosis, which is very harmful. So, a scaffold that allows to maintain, as viable MSCs at the site of the bone injury should always be considered.”

  1. The authors had better explain relationship and differences between this paper and previous papers such as [16] and [35]. How position was this paper in your works? What was new?

R: Both references cited are studies carried out by our research group that explains the methodological similarity employed. The main objective is to find the better application of Fibrin Biopolymers as a scaffold for stem cells in bone injury models, in order to carry out tests in human patients in the near future.

Gasparotto et al., in “A New Fibrin Sealant as a Three-Dimensional Scaffold Candidate for Mesenchymal Stem Cells” [15] was pioneered and evaluated for the first time in vitro effects of Fibrin Biopolymer as a three-dimensional scaffold for mesenchymal stem cells. The results obtained by survival and differentiate abilities of cells encouraged the group to seek its application in animal models.

Orsi et al., in “A Unique Heterologous Fibrin Sealant (HFS) as a Candidate Biological Scaffold for Mesenchymal Stem Cells in Osteoporotic Rats” [36] evaluated the treatment of critical defects in femurs of osteoporotic rats. The animals were observed just for a short period, preventing complete bone regeneration. In our current study, we observed the animals for 42 days, which after pilot tests proved to be sufficient. In addition, we evaluated animals without any healing problem, as Orsi et al., that evaluated animals with osteoporosis, which certainly interferes with bone supply.

Reviewer 3 Report

The manuscript of Creste et al. is devoted to the use of a fibrin-based biopolymer as matrix for mesenchymal stem cells taking part in the bone regeneration. The manuscript is well-planned and concisely written. I believe it to be suitable for publication in Materials after minor revision. 

Some comments are given below:

Could the authors give some comments on functional capabilities of a regenerated bone? Did they make any animal functional tests? Since the restoration of the bone condition is demonstrated, it would be interesting to know how the regenerated bone works.

The introduction section is too consise. One-phrase paragraphs look weird. 

The English deserves thorough proofreading and revision. There are also some typos in the text as well as some incorrect referencing. 

Author Response

Dear reviewer. Thank you very much for all the considerations. We are answering each of them below.

REVIEW REPORT 3

Comments and Suggestions for Authors

The manuscript of Creste et al. is devoted to the use of a fibrin-based biopolymer as matrix for mesenchymal stem cells taking part in the bone regeneration. The manuscript is well-planned and concisely written. I believe it to be suitable for publication in Materials after minor revision. 

Some comments are given below:

Could the authors give some comments on functional capabilities of a regenerated bone? Did they make any animal functional tests? Since the restoration of the bone condition is demonstrated, it would be interesting to know how the regenerated bone works.

R: No type of mechanical and / or functional test was performed on the regenerated bone because it was not the purpose of the study. The reviewer's comment was very pertinent and will certainly be the objective of a future study. We thank. It was possible to observe that the animals did not show signs of pain or infection at the lesion site and, in addition, they moved normally, from the immediate postoperative period to the observation period.

The introduction section is too consise. One-phrase paragraphs look weird. 

R: The introduction was reviewed.

The English deserves thorough proofreading and revision. There are also some typos in the text as well as some incorrect referencing. 

R: The English was reviewed and some typos were corrected
